



# Dynamics of Rare Earths and associated major and trace elements during Douglas-fir (*Pseudotsuga menziesii*) and European beech (*Fagus sylvatica* L.) litter degradation.

Alessandro Montemagno[1,3], Christophe Hissler[1], Victor Bense[3], Adriaan J. Teuling[3], Johanna Ziebel[2], Laurent Pfister[1]

[1]CATchment and ecohydrology research group (CAT/ENVISION/ERIN), Luxembourg Institute of Science and Technology, Belvaux, 4408, Luxembourg

[2]Biotechnologies and Environmental Analytics Platform (BEAP/ERIN), Luxembourg Institute of Science and
Technology, Belvaux, 4408, Luxembourg

[3]Department of Environmental Sciences, subdivision Hydrology and Quantitative Water Management (HWQM), Wageningen University and Research, Droevendaalsesteeg 4, Wageningen, 6708 PB, The Netherlands

*Correspondence:* Alessandro Montemagno (alessandro.montemagno@list.lu), Christophe Hissler (christophe.hissler@list.lu)

**Abstract.** Given the diverse physico-chemical properties of elements, we hypothesize that their incoherent distribution across the leaf tissues, combined with the distinct resistance to degradation that each tissue exhibits, leads to distinct turnover rates between elements. Moreover, litter layers of different ages produce diverse chemical signatures in solution during the wet degradation. To verify our hypothesis, Na, K, Mg, Mn, Ca, Pb, Al and Fe were analysed together with the Rare Earth Elements (REE) in the solid fractions and in the respective
leachates of fresh leaves and different humus layers of two forested soils developed under *Pseudotsuga menziesii* and *Fagus sylvatica* L. trees. The results from the leaching experiment were also compared to the *in situ* REE composition of the soil solutions to clarify the impact that the litter degradation processes may have on soil solution chemical compositions.

Our results clearly show that REE, Al, Fe and Pb were preferentially retained in the solid litter material, in
comparison to the other cations, and that their concentrations increased over time during the litter degradation. Accordingly, different litter fractions produced different yields of elements and REE patterns in the leachates, indicating that the tree species and the age of the litter play a role in the chemical release during the degradation. In particular, the evolution of the REE patterns according to the age of the litter layers allowed us to deliver new findings on REE fractionation and mobilization during litter degradation. In particular, the $La_N/Yb_N$ ratio
highlights differences in litter degradation intensity between both tree species, which was not shown with major cations. We finally showed the primary control effect that litter degradation can have on the REE composition of the soil solution, which presents a positive Ce anomaly associated with the dissolution and/or transportation of Ce-enriched $MnO_2$ particles accumulated onto the surface of the old litter due to white fungi activity. Similar MREE and HREE enrichments were also found in the leachates and the soil solution, probably due to their higher
affinity to the organic acids, which represent the primary products from the organic matter degradation.





## 1 Introduction

Nutrient cycling is key to forest ecosystem sustainability and productivity, especially in sites characterized by low fertility or degraded soils. There are three types of nutrient cycles, which relate to geochemical, biochemical or biogeochemical processes (Morris, 2004). The geochemical cycle encompasses all processes inherent to the

introduction or removal of nutrients – excluding any kind of biological activity (e.g., input from aerosols, leaching of nutrients from rocks and their removal from the system through runoff). The biochemical cycle refers to processes involved in the transport and retention of nutrients inside the trees (such as the withdrawal of specific nutrients from leaves before senescence). The biogeochemical cycle encompasses the processes that occur outside of the trees and lead to the degradation of the organic waste material (such as exudates of leaves and

stems, dead leaves and branches or even a whole tree) into its primary components and therefore to the release of nutrients in a form that is reusable by trees (Morris, 2004). Organic matter degradation, which represents part of the biogeochemical cycle, is a major contributor to the nutrient stock available to trees in forest ecosystems (Staaf, 1980; Guo and Sims, 1999 and references therein; Chadwick *et al*., 1999; J.M. Pacyna, 2008; M. P. Krishna and M. Mohan, 2017) and in this context, litter degradation is known to play a key role in the

replenishment of the nutritional pools of forests (Tagliavini *et al*, 2007). Nutrient release from litter is possibly regulated by various biotic and abiotic factors: the temperature, abundance of precipitation, species of decomposers (including the microfauna and microorganism communities), litter composition and chemical structure of its components are all factors that regulate the degradation rate and therefore the recirculation of the elements in a forest (Krishna and Mohan, 2017 and references therein).

55       Trees absorb many nutrients to supply metabolic demands for growth, the immune system and reproduction but, at the same time, unnecessary elements, such as toxic metals, can also be absorbed and "trapped" in specific tree's compartments (Gomez *et al.*, 2018). Indeed, once the elements are absorbed, they are distributed within different tissues, depending on their metabolic role or on their affinities with various compounds (Shan *et al*., 2003; Ding *et al*., 2005; Brioschi *et al*, 2013; Page and Feller, 2015; Ming Yuan *et al*.,

2017). This distribution could play a key role in processes involved in element turnovers in forest ecosystems especially during litter degradation, which potentially leads to a preferential release into the environment of some elements rather than others, depending on the substances to which they are bound. Litter degradation, indeed, would preferably promote the release of elements from more labile fractions making them available for tree uptake during the first stages of the degradation (Swift *et al*., 1979), while pools of elements that are trapped

inside the most refractory tissues would remain unavailable for longer time spans.

The study of the biogeochemical processes involved in the distribution of the different classes of elements (toxic and nutrients) among the various leaf's tissues during the growth period, and their fractionation during the degradation of the litter is of crucial importance for a better understanding (and forecast) of the dynamics of the aforementioned classes of elements in forest ecosystems. To investigate such processes, Rare

Earth Elements (REE) are interesting candidates due to their recognized use as tracers of geochemical processes, existing knowledge of their partitioning in plant tissues and recent studies related to their ecotoxicology as emerging pollutants. Indeed, knowledge of the processes involved in REE dynamics during litter degradation is of importance to environmental and social matters due to the increase in their environmental concentrations linked to their extraction in mining areas and exploitation in modern technologies (Xiaofei Li *et al*., 2013; Kyung

Taek Rim *et al*., 2013 and reference therein). REE are a group of elements composed of lanthanides (from ${}^{57}$La





to $^{71}$Lu) and $^{39}$Y. Apart from Y, the other REE are usually divided into light (LREE: La to Nd), middle (MREE: Sm to Tb) and heavy (HREE: Dy to Lu) according to their atomic weight. To highlight the geochemical behaviour characteristics of the REE, it is convenient to consider the normalized concentrations rather than their absolute concentrations as is usually the case for other chemical elements. From the trends of the normalized

concentrations (or patterns), it is then possible to determine the REE serial behaviour. These patterns can exhibit so-called "anomalies", which represent an enrichment (positive anomalies) or depletion (negative anomalies) in certain elements of the series or also a fractionation occurring among the three groups of elements that are intimately linked to the environmental conditions. The characteristics of the patterns then, allow us to establish biogeochemical processes occurring in the system. REE have already proven to be among the best-suited tracers

for investigating Critical Zone processes such as: the origin of solid and dissolved load transported by stream (Aubert *et al*., 2001; Hissler *et al*., 2015a); metal adsorption in organic matter (Schijf and Zoll, 2011) and in bacterial cell walls (Takahashi *et al*., 2005 and 2010); characterization of water-rock interaction and regolith weathering processes (Bau, 1996; Aubert *et al*., 2001; Stille *et al*., 2006, Ma *et al*., 2011, Hissler *et al*, 2015b; Jin *et al*., 2017; Laveuf and Cornu, 2009; Moragues-Quiroga *et al*., 2017, Vázquez-Ortega *et al*., 2015 and 2016);

as an indicator for atmospheric dust composition in leaves (Censi *et al*., 2017); or wastewater spillage in freshwaters (Merschel *et al*. 2015; Hissler *et al*., 2016); and metal mobilization and fractionation in the soil-plant continuum (Liang *et al*., 2005 and 2008; Censi *et al*., 2014a; Cheng *et al*., 2014; Ding *et al*., 2006; Srmhi *et al*., 2009).

Despite the large number of REE studies on plant tissues (Fu et al. 1998; Wyttenbach et al. 1998; Wei
et al. 2001; Han et al. 2005; Ding et al. 2005, 2006; Brioschi et al. 2013; Censi et al. 2014; Zaharescu et al. 2017), their dynamics during litter degradation is still scarcely known, since research has mainly focused on changes in the concentrations of elements according to the total mass loss of litter material during the duration of the experiment (Tyler, 2004; Brun *et al*., 2008; Gautam *et al*., 2020). To the best of our knowledge, no studies regarding the processes involved in the regulation of such dynamics have been carried out. Our aim is to elucidate
which processes may control the release and retention of REE in relation to major cations and other trace elements during the wet litter degradation – with a secondary focus on the qualitative impact of litter degradation on the REE patterns of soil solutions. We believe that REE environmental behaviour and its capacity to accurately trace biogeochemical processes add value to our understanding of nutrient cycles and in particular, could more precisely inform us about the processes that control the litter degradation stages and the release of nutrients in
forest ecosystems. Since the different leaf tissues (across which leaves distribute the uptaken elements during the living period) exhibit distinct resistance to degradation, we hypothesize that during litter decay the combination of such a distribution process with the different levels of degradation resistance of the tissues will lead to distinct turnover rates among the elements. Consequently, this leads to a specific chemical release into the environment depending on the degradation stage of litter and the different tissues among which the elements
are distributed. To test our hypothesis, we designed a field experiment in the forested Weierbach experimental catchment (Hissler *et al*., 2021), relying on a series of biogeochemical tracers – including concentrations of Na, Mg, K, Ca, Mn, Fe, Pb, Al and REE measured in fresh leaves and different litter fractions (sorted by degradation degree) of Douglas-fir (*Pseudotsuga menziesii)* and European beech (*Fagus Sylvatica* L.) grown on the same soil. Moreover, we carried out leaching experiments on these samples with ultrapure water (MilliQ) in order to
observe how the different fractions of litter can contribute to element release and sequestration and how this





release may affect soil solution chemistry. Finally, we compared the leaching experiment results to the chemical composition of soil solutions collected from the two tree stands. If our hypothesis is confirmed, we expect the older litter fractions to show an enrichment in specific elements, which would be linked to their distribution in the most refractory tissues, limiting their release into the environment during the degradation. Subsequently, we

expect that the solutions obtained through the leaching experiment (hereafter called "leachates") will exhibit different chemical signatures according to the degree of degradation of the samples. On the contrary, if different litter fractions showed similar chemical compositions and similar chemical release during the leaching experiment, it would imply that the distribution of the elements among the tissues is coherent and, therefore, the degradation stage of the litter does not affect the chemical release during the litter decay.

**2 Materials and methods**

**2.1 Study site**

We selected two experimental plots in the Weierbach catchment located in the Luxembourg Ardennes Massif, which have been monitored for ecohydrological purposes since 2012 (Hissler *et al.*, 2021). The "Be" profile (BeP) shows a deciduous cover of European beech(*Fagus sylvatica L.*), while the "Do" profile (DoP) is covered

with Douglas-fir (*Pseudotsuga menziesii*). The altitude ranges from 450 to 500 m a.s.l. and the geological substratum consists of Devonian metamorphic slates covered by 70 to 100 cm of Pleistocene Periglacial Slope Deposit (PPSD) composed of past loamy aeolian deposition (Moragues-Quiroga *et al.*, 2017). The soil, which is developing below a Hemimoder type of humus (Jabiol *et al*., 2013) in the first 50 cm of the PPSD, presents homogenous properties all over the catchment. It is at an early formation stage and classified as dystric cambisol

according to the World Reference Base for soil resources (Juilleret *et al*, 2016).

**2.2 Sampling and preparation**

The different humus layers (i.e. fresh leaves and needles, as well as new and old litter) were collected on the same sampling day in May 2019 at both beech and Douglas-fir stands.

    Fresh leaves (beech) and needles (Douglas-fir) (hereafter referred to as FL) were collected from 10

adult trees randomly selected per plot. The leaves were taken from different branches, accessible from the ground, at various heights and radial directions. All leaves were aggregated together in one sample per plot and stored in clean polypropylene bags. The litter material was collected from five different locations within an area of 500 m$^2$ of each experimental plot using a 25x25cm metallic frame and avoiding contamination by soil particles. During the collection, the different fractions of litter were sorted according to their degradation degree

(Fig. 1) and stored in polypropylene bags.

    In BeP, three litter fractions were identified: the new litter (OLn - unprocessed, unfragmented, light-brownish coloured), the old litter (OLv - slightly altered, bleached and softened, discoloured or dark-brownish coloured) and the fragmented litter (OF - partially decomposed and fragmented, grey-black coloured). For DoP only two fractions stood out: the new litter (OLn - unprocessed, unfragmented, light-brownish coloured) and the

old litter (OLv - slightly altered, bleached and softened, grey-black coloured), whereas the fragmented litter layer was not sufficiently developed to be representative as a humus layer and was not considered in this study.

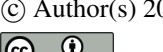



Douglas-fir (*Pseudotsuga menziesii*) samples

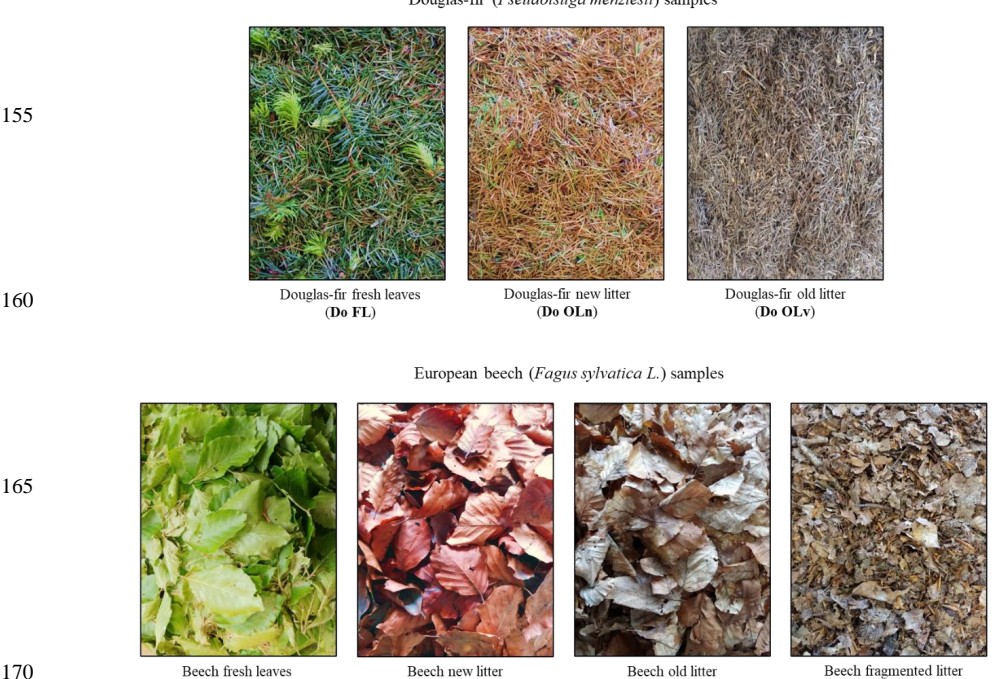


Douglas-fir fresh leaves
**(Do FL)**     Douglas-fir new litter
**(Do OLn)**     Douglas-fir old litter
**(Do OLv)**

European beech (*Fagus sylvatica L.*) samples


Beech fresh leaves
**(Be FL)**     Beech new litter
**(Be OLn)**     Beech old litter
**(Be OLv)**     Beech fragmented litter
**(Be OF)**

**Figure 1**: Fresh leaves and litter of European beech and Douglas-fir collected in the Weierbach Catchment and sorted by

All fresh leaves and humus material were cleaned with a strong air flux and with a brush avoiding the use of water in order not to disturb the chemical signature and the biotic communities existing on the samples' surfaces. The samples were then dried in the oven at 40 °C and homogenized.

Two aliquots were taken to perform the total chemical composition and the leaching experiment, respectively. The leaves and litter aliquots for the total chemical composition were preliminarily reduced to a fine powder and incinerated in closed ceramic cups at 550 °C before mineralization in order to destroy OM and
to pre-concentrate trace elements. After the burning, the residual ash was digested using aqua regia (ultrapure concentrated $HNO_3$ and HCl) in a microwave-assisted oven (Anton Paar Multiwave PRO) and stored at 4°C before analysis. For the leaching experiment, 2L high-density polyethylene bottles were filled with the aliquots of leaves and litter fractions. 1L of ultrapure water was then added. There were two reasons for putting as much material as possible into the bottles: first, part of the samples had to be above the water level in order to enhance
the aerobic degradation; and second, we wanted to be sure that the release of elements during the experiment period was abundant enough to be detectable with the instrumentations. The bottles were left with the cup partially open in order to allow gas exchanges to encourage bacterial and fungi activity. Samples were agitated for 1 hour/day for 7 days in an automatic vertical agitator (GFL type 3040) set at minimum speed to enhance the leaching process from all samples while avoiding further fragmentation. After one week, the leachates were
separated from the solid material with a nylon sieve, filtered at 0.2 µm and acidified using $HNO_3$ (1% in volume). 50 ml aliquot from each leachate was evaporated in Savillex PFA vessels placed on a hot plate allowing the





precipitation of the content. The residue was then sequentially mineralized with HF, HCl and HNO₃. After evaporation, the residue was then dissolved in HNO₃ (1% in volume) and samples were stored at 4°C before the analysis. Aliquots were also taken for DOC and pH measurement.

The atmospheric dust was collected at BeP using a modified polypropylene version of the passive SIGMA-2 collectors produced by the German Meteorological Service in Freiburg, Germany (VDI 3787, 2010). The SIGMA-2 passive sampler allows the sampling of coarse atmospheric particles (>2.5µm). The special construction of the collector (Grobéty *et al.*, 2010) allows the sampling of coarse (>2.5µm) atmospheric particles at 1m above the ground. The atmospheric dust represents an integrated sample exposed to the atmospheric

deposition from September 2018 to May 2019. The particles were collected and stored in an acid-cleaned Teflon vessel during the whole exposure period and stored in a clean desiccator in the laboratory until its preparation before the analysis. After precise weighing of the atmospheric dust collected (2.3 mg), the sample was sequentially mineralized using HNO₃, HF and HClO₄ in a digestion procedure using concentrate ultrapure acids. The acids were then evaporated, and the residue was dissolved in a solution of HNO3 (1% in volume) and stored

at 4°C before the analysis.

Thanks to the bi-weekly monitoring in place at the Weierbach catchment since 2009 (Hissler *et al.*, 2021), we can rely on the chemical composition of soil solutions collected between 2012 and 2014 at the two sampling locations at 20, 40 and 60 cm depths. The sampling was performed using Teflon/quartz suction cups (SDEC, Reignac-sur-Indre, France) connected to 2L-Nalgene flasks under a vacuum of 0.8 bar.

**2.3 Sample analysis**

The concentrations of major cations and trace elements in all samples were analysed via Inductively Coupled Plasma - Mass Spectrometry (Agilent 7900). The limit of quantification (LoQ) for the different analyzed elements are reported in Table SI-1.

DOC was measured via a Teledyne Tekmar® Torch Combustion Analyser and pH with SenTix® 940 WTW.

**2.4 REE normalization and anomaly calculations**

All data presented in the following sections were normalized to the local atmospheric deposition (Table SI-2). This allowed us to compare our samples to a local reference for REE. As atmospheric dust is an important input of cations and nutrients (Reynolds *et al.*, 2006; Lequy *et al.*, 2012) we also expected it to be important in terms of REE input in forest ecosystems. Moreover, with this normalization, we could directly differentiate the

vegetation contribution to the REE patterns observed for the litter samples.

Gadolinium (Gd), Europium (Eu) and Cerium (Ce) anomalies were calculated from equations 1 to 3, respectively:

$$Gd/Gd^* = Gd_N/(0.33 \times Sm_N + 0.67 \times Tb_N) \qquad (1)$$
$$Eu/Eu^* = Eu_N/(0.5 \times Gd_N + 0.5 \times Sm_N) \qquad (2)$$
$$Ce/Ce^* = Ce_N/(0.5 \times La_N + 0.5 \times Pr_N) \qquad (3)$$

with $La_N$, $Ce_N$, $Pr_N$, $Sm_N$, $Eu_N$, $Gd_N$ and $Tb_N$ indicating the REE concentrations normalized to the local atmospheric dust.



## 3 Results

### 3.1 Chemical composition of the fresh leaves and litter bulk samples

Concentrations of REE in fresh leaves and litter are reported in Table SI-3. When normalized to the REE concentrations of the local atmospheric deposition (hereafter referred to as "dust"), the REE patterns of bulk fresh leaves and litter material show some similarities between the two tree species. The REE concentrations increase with the age of the litter, with the lowest concentration in fresh leaves and the highest in the most degraded litter layers (Fig. 2a-b and Table SI-3). Additionally, both tree species present a depletion in HREE according to LREE and MREE (Fig. 3a-b). Y appears to behave coherently during the litter degradation, as illustrated by the evolution of the Y/Ho ratios. Indeed, the samples show a Y enrichment, according to the dust (Y/Ho = 25.93), that is higher in the fresh leaves, with Y/Ho ratios equal to 34.34 and 34.3 for Do FL and Be FL, respectively. This ratio decreases progressively with the age of the litter, with the lowest values in the oldest litter fractions (23.58 and 25.43 in Do OLv and Be OF, respectively).

**Figure 2:** Log$_{10}$ of the concentrations of the studied elements in fresh leaves and different litter fractions of (a) Douglas-fir and (b) European beech and element yields studied (% of the total mass) for the leaching experiment of of (a) Douglas-fir and (b) European beech fresh leaves and litter samples.





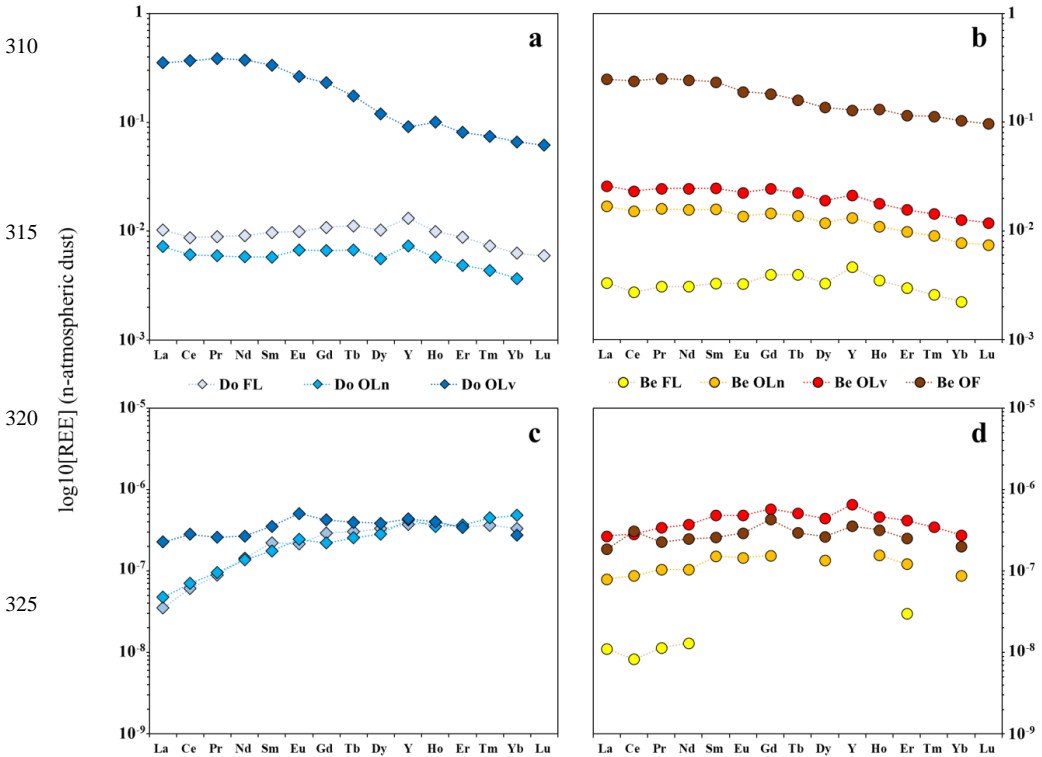

**Figure 3**: Patterns of Rare Earth Elements concentrations in samples of fresh leaves and litter of Douglas-fir (**a**) and beech (**b**); patterns of Rare Earth Elements concentrations in leachates of fresh leaves and litter of Douglas-fir (**c**) and beech (**d**). REE concentrations were normalized by the values in the local atmospheric dust.

The REE concentration and the LREE enrichment also increase in line with the litter degradation stages but with significant differences between the Douglas-fir and the European beech. For the Douglas-fir samples, the REE total concentrations decrease from the fresh material to the Do OLn litter layer before drastically increasing at the Do OLv sample. The total REE concentrations are 0.78, 0.51 and 25.3 µg g$^{-1}$ in Do FL, Do OLn and Do OLv, respectively. Here, the fractionations between the REE groups are not visible between the fresh leaves and the Do OLn litter layer. However, the fractionation between LREE, MREE and HREE in Do OLv are the highest for all the samples considered in this study, having La$_N$/Yb$_N$ and La$_N$/Gd$_N$ and Gd$_N$/Yb$_N$ ratios of 5.35 and 1.53 and 3.5, respectively. For the European beech, the REE total concentration in the fresh leaves is the lowest for all leaves and litter samples (Fig. 3b). The total concentration increases progressively at each degradation stage to reach its maximum value of 17.7 µg g$^{-1}$ for the Be OF litter layer. The LREE enrichment increases as illustrated by the La$_N$/Yb$_N$ ratios, which progressively evolve from 1.49 in the fresh leaves to 2.40 in the Be OF layer. The degradation of the beech litter also leads to an MREE enrichment as illustrated by the La$_N$/Gd$_N$ ratio evolving from 0.84 in Be FL to 1.16 in Be OLn and finally 1.36 in Be OF.

The major elements and metals can be sorted into three different groups according to the evolution of their concentrations in the bulk samples of the different litter layers (Fig. 2a-b). This classification stays coherent for





the two tree species. Na, K and Mg have their highest concentrations in the fresh leaves and the older litter layers. Ca and Mn present a progressive increase of their concentration for the European beech, whereas they are less concentrated in the oldest litter layer of Douglas-fir in comparison to Do FL and Do OLn. The concentrations of the other trivalent metals (Fe, Al) evolve similarly to the REE with a progressive increase from the fresh leaves to the OL litters and a significant enrichment in the oldest litter layer for both species (Do OLV and Be OF). In

contrast to the European beech, Douglas-fir fresh leaves present metals concentrations as high as in the Do OLn litter sample.

**3.2 Chemical composition, pH and DOC content of leachates**

The leaching experiment led to similar REE concentration ranges between the two tree species, except for the beech fresh leaves, which are one order of magnitude less concentrated (Table SI-4). The leachate patterns present

strong differences to the REE characteristics of the bulk leaves and litter material and in between the two tree species.

The total REE concentrations of the Douglas-fir leachates are similar in Do FL and Do OLn samples and higher in the leachate of the Do OLv sample. As the HREE show very similar concentrations in the three leachates, the difference is mainly related to the concentration of LREE, as shown by the dust-normalized REE patterns (Fig. 3-

c). Indeed, LREE are similarly depleted in the leachates of the two younger samples, with $La_N/Yb_N$ ratios of 0.10, while Do OLv presents a $La_N/Yb_N$ ratio equal to 0.82. Noticeable are the significant Eu positive anomalies (Eu/Eu*) of 1.23 and 1.31 observed in the leachates of Do OLn and Do OLv, respectively. In Do OLv a slight positive Ce anomaly (Ce/Ce*) of 1.16 is also observed.

        The REE concentrations in the leachates of beech samples increase from the fresh leaves to the highest

stages of litter degradation but are higher for the Be OLv material. The dust-normalized REE patterns of beech leachates have an MREE enrichment in comparison to LREE and HREE where Gd shows the highest concentrations. The patterns of Be OLn and Be OLv leachates present very similar characteristics as indicated by their $La_N/Gd_N$ ratios (0.46 and 0.43, respectively), $Gd_N/Yb_N$ ratios (2.10 and 2.14, respectively) and the absence of any anomaly, whereas the Be OF leachate presents significant Ce and Gd positive anomalies (Ce/Ce*=1.49 and

Gd/Gd*=1.52). In contrast, the leachate of the fresh beech leaves presents a Ce negative anomaly (Ce/Ce*=0.74).

        Similar trends in the percentage of elements leached from the material of both tree species were observed. The percentage of leaching of the studied elements can be classified according to their valence Na, K > Mg, Mn > Ca, Pb > Al, Fe, REE, with the trivalent elements being less leached. However, some differences can be highlighted between beech and Douglas-fir.

The Douglas-fir material released a higher quantity of Na, Mg, Mn, Pb, Al, Fe, MREE and HREE compared to the beech. Major elements and Mn are preferentially released compared to trivalent metals and Pb, as shown in Figure 2c. The highest percentage of element release is observed from the Do OLn sample with Mn having similar values in Do OLn-L and Do OLv-L. An exception is made for Ca, which presents similar release percentages as those of the trivalent metals during the first stages of degradation and which show the highest

release from the Do OLv fraction.

        In beech leachates, the elements show similar release trends as for Douglas-fir but the highest percentages of release for all elements are shown in Be OLv (Figure 2d) with Na and K having analogous values in Be OLn and Be OLv. In Be FL and Be OLn leachates, all the elements show lower release values than in Douglas-fir





samples. Trivalent metals and Pb, similarly to those of Douglas-fir, show a low release from the solid material

during the experiment (in the case of Al, the release from fresh leaves is below the limit of quantification, as well as for many REE as shown in Table SI-4 and SI-5).

The Y/Ho ratios of the leachates of Douglas-fir litter can explain the appearance of a small Y depletion in the oldest litter sample. The leachate of Do OLn, which represents the stage of degradation that brings about the formation of the OLv fraction, indeed shows a Y/Ho ratio equal to 31.29 indicating a preferential release of Y

(when compared to the neighbour) that leads to a lower-than-atmospheric dust value in the Do OLv sample (Y/Ho = 23.58 in Do OLv and Y/Ho = 25.93 in atmospheric dust).

In beech samples, we can observe a similar behaviour with values that are slightly higher. The Y/Ho ratio in the Be OLv leachate (Y/Ho = 36.59) justifies the absence of Y enrichment in the Be OF fraction, which instead shows a dust-like ratio (Y/Ho = 25.43).

The highest DOC concentrations (Table SI-4) were measured in the Douglas-fir leachates with values ranging from 10.39 mg $L^{-1}$ in Do OLv to 29.37 mg $L^{-1}$ in Do OLn, while beech leachates showed concentrations from 5.91 mg $L^{-1}$ in Be OLn to 14.68 mg $L^{-1}$ in Be OLv. Note that for both species, the highest DOC concentrations were measured in leachates in the second-to-last degradation stages with a significant decrease in the oldest fractions.

The pH appears to be inversely proportional to the DOC concentrations. Indeed, for the Douglas-fir leachates, the most acidic pH was found in Do OLn, which showed a pH = 4.26 (Table SI-4), while the pH of Do OLv was the highest with a value equal to 5.03. In beech leachates, the lowest pH was measured in Be OLv (pH = 4.07) and the highest in Be OLn (pH = 5.39).

### 3.3 Average REE in soil solutions

The average REE concentrations in soil solutions collected between 2012 and 2014 are reported in Table SI-6. The REE total concentrations differed in one order of magnitude and were lower under the Douglas-fir stand at 20 and 40 cm depth (ΣREE=0.88 μg $L^{-1}$ and 0.92 μg $L^{-1}$ in Do SS20 and Do SS40, respectively), whereas the highest concentration was observed in beech samples at 40 cm depth (ΣREE=6.70 μg $L^{-1}$ in Be SS40).

The dust-normalized REE patterns show an MREE enrichment for all soil solutions and Ce positive

anomalies at 20 and 40 cm depth. Eu has the highest MREE concentration in the soil solutions under the Douglas-fir, whereas Gd is most concentrated in beech soil solutions. Do SS at 20 and 40 cm depth exhibit positive Ce anomalies (Ce/Ce* = 1.14 and 1.21, respectively) that disappear at 60 cm. Moreover, Douglas-fir samples also show an LREE depletion as indicated by the $La_N/Yb_N$ ratios, which range from 0.50 to 0.66 in Do SS40 and Do SS60, respectively.

Under the beech, the Ce positive anomaly is higher at 20 cm and decreases with depth until it becomes negative in soil solutions at 60 cm depth (Ce/Ce* = 1.39, 1.20 and 0.82 in Be SS20, Be SS40 and Be SS60, respectively), whereas the LREE show a consistent depletion at 20 cm depth ($La_N/Yb_N$ = 0.56) and a slight one at 60 cm depth ($La_N/Yb_N$ = 0.85).





## 4 Discussion

### 4.1 REE fractionation during litter degradation: similarities and differences with the other elements

Similarities in the shapes of REE patterns for litter from the two tree species can be observed among the different degradation stages (Fig. 3a-b). Patterns of litter material are characterized by an increasing LREE enrichment over time, which is more evident in the Douglas-fir samples as suggested by the comparison of the $La_N/Yb_N$ ratios in the Do OLv and Be OF samples ($La_N/Yb_N$=5.35 and $La_N/Yb_N$=2.40, respectively).

The leaching yields (Fig. 2c-d) clearly illustrate that during the leaching experiment, REE are preferentially leached following the HREE>MREE>LREE order. This justifies the decreasing trends (from La to Lu) of the REE patterns of litter fractions for both tree species and demonstrates the tendency of the litter to retain LREE rather than the other elements of the lanthanide series.

     The leaching experiment also showed that REE, together with the other trivalent metals (Fe and Al) and 435 Pb, are preferentially held inside the solid material, while the other major elements and Mn are more easily released during litter degradation. This behaviour suggests a preferential fractionation of the studied nutrients (Na, K, Mg and Mn) into more labile tissues, probably due to their role in the metabolic functioning of the tissues of living leaves (Alejandro *et al.*, 2020; Sardans and Peñuelas, 2021; Shaul, 2002; Maathuis, 2013). This partitioning results in a concentration decrease of these elements during litter degradation, while Fe, Al, Pb and REE tend to remain 440 embedded inside the most refractory tissues - increasing their concentration over time in the remnant litter material. In our experiment, all elements behave coherently according to their oxidation number (Fig. 2c-d). Monovalent elements (Na, K) are more likely to be released than the divalent elements (Mg, Mn), while trivalent metals (Al, Fe, REE) tend to stay more tightly bound to the solid residual fraction. Exceptions can be seen for Ca in the Douglas-fir samples and for Pb in the oldest litter fractions of both tree species.

Given that REE are preferentially retained in the recalcitrant tissues together with the other trivalent metals and Pb, the higher yields for these elements obtained from Douglas-fir samples during the leaching experiment might suggest a faster degradation process for the Douglas-fir litter when compared to the beech litter. The fact that LREE are preferentially retained in the litter of both tree species can be explained by referring to existing literature and by normalising the REE concentrations in the leachates to the ones of the respective litter 450 material (Fig. 4).

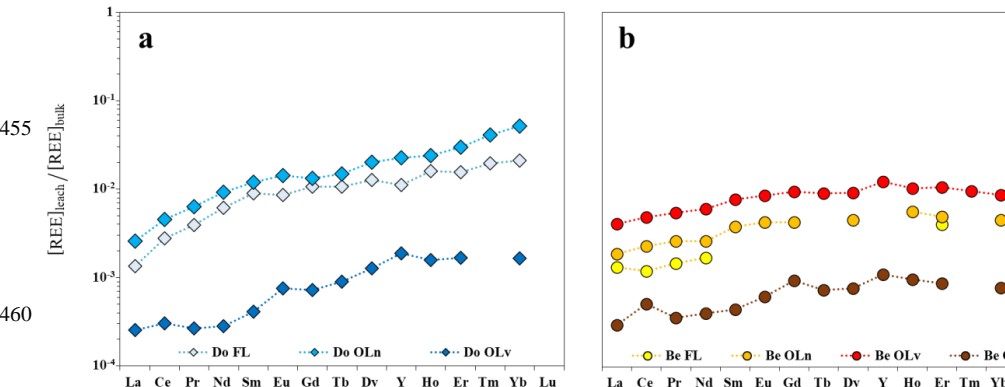

**Figure 4.** Patterns of the REE in leachates normalized to the REE concentrations in leaves and litter for Douglas-fir (**a**) and Beech (**b**) samples.





The patterns of REE in Douglas-fir samples show an HREE enrichment when compared to the other elements of the series ($0.05 \leq La_N/Yb_N \leq 0.15$ and $0.25 \leq Gd_N/Yb_N \leq 0.51$), indicating a preferential release of the heavy REE to the solution. In the European beech samples, the patterns are smoother between MREE and HREE ($0.94 \leq Gd_N/Yb_N \leq 1.21$), while they conserve the depletion in LREE when compared to the other groups ($0.32 \leq La_N/Gd_N \leq 0.44$ and $0.38 \leq La_N/Yb_N \leq 0.48$).

In general, these patterns show a progressive increase from La to Lu that mirrors the trend of the stability constants of REE complexes with malate, EDTA, humic and fulvic acids (Suzuki *et al.*, 1980; Ding *et al.*, 2006; Pourret *et al.*, 2007; Marsac *et al.*, 2010; Sonke and Salters, 2005). Humic acids, fulvic acids and malic acid are indeed primary products of organic matter degradation (Adeleke *et al.*, 2017 and reference therein) and they have been reported to have metal chelating properties. The increasing release of REE from La to Lu in our leachates is likely due to an increase in the specific affinity - proceeding along the lanthanide series - of these elements towards the degradation products present in the aqueous solution (Schijf and Zoll, 2011). This suggests that the nephelauxetic effect (Juranic, 1988; Tchougréeff and Dronskowski, 2009) plays a key role in the mechanism of REE complexation with the dissolved organic ligands during the wet degradation of litter and is in line with the results shown by Sonke and Salters (2006), which experimentally demonstrated that the lanthanide contraction is responsible for a gradual increase in the complexation strength with humic substances when decreasing the ionic radius. The bond strength, indeed, increases with increasing ionic potential Z/r.

Differences in the atmospheric dust-normalized REE patterns of leachates between the two tree species can be explained by the nature of the ligands present in the solutions. Tang and Johannesson (2010) showed that the REE complexation with organic ligands in natural waters would produce patterns enriched in HREE when the majority of the ligands in solutions are represented by low molecular weight-dissolved organic compounds (such for instance citric, oxalic, malic, succinic, malonic and maleic acids) while a preponderant presence of heavy molecular weight-dissolved organic compounds (such as humic acid and fulvic acids) would produce an enrichment in MREE. Therefore, the results obtained indicate that the differences between the patterns of Douglas-fir and European beech leachates are likely related to the production of different classes of organic acids during the degradation of the samples.

Al, Fe, Pb and REE behaviours are coherent during litter degradation for both tree species, as their concentrations progressively increase towards the oldest litter fractions (Table SI-3 and Fig. 2a-b). We can explain such an accumulation in the oldest litter fractions by the binding with lignins. Lignins constitute the most degradation-resistant compounds in leaves. Their resilience lets these metals persist longer in the organic material than carbohydrates, lipids, proteins, hemicellulose and cellulose, which can degrade at a faster rate (Rahman *et al.*, 2013). Lignins are a class of organic polymers that have many functions in vascular plants. They provide structural support, improve cellular adhesion, enhance water transport and defence towards pathogens and are mainly situated in the cell walls of vascular and support tissues (Weng and Chapple, 2010; Leisola et al., 2012; Labeeuw *et al.,* 2015). Moreover, the chemical structure of these molecules exerts a strong control on litter decay rates (Talbot *et al.*, 2012). Increased lignin concentrations inhibit biological activity and linearly increase photo-degradation due to its wide spectrum of absorbance (Austin and Ballaré, 2010; Cogulet *et al.*, 2016).

While Fe, Al and Pb toxicity in plants is well-known (Imadi *et al.*, 2016; Bienfait, 1989; Woolhouse, 1983, Rout *et al.*, 2001; Pallavi Sharma and Rama Shanker Dubey, 2005; Pourrut *et al.*, 2011; Singh *et al.*, 2017; Miroslav Nikolic and Jelena Pavlovic, 2018), the toxicity of REE is not yet widely studied as their micropollutant





nature has only recently emerged (Gwenzi *et al.*, 2018). However, the toxicity of REE in plants is far beyond the scope of this work, we limit ourselves to mentioning that researchers observed REE displaying redox-related

toxicity mechanisms (Hassan Ragab El-Ramady, 2010; Pagano *et al.*, 2015) and we assume that plants can trap REE in lignified tissues as a defence to avoid the toxicity-related events with the same mechanism adopted for other potentially toxic metals, such as Pb and Al. Therefore, we propose that lignins constrain the REE in the oldest litter fractions during the degradation of the leaves. Given the high affinity of such metals for oxygen, the absorption operated by lignins through the binding with the oxygen-bearing functional groups (such as phenolic,

hydroxyl) may be the mechanism involved and would explain the accumulation of these metals in the oldest litter fractions. Therefore, during the living cycle of leaves, lignins are able to sequestrate the elements that show higher affinity for the exposed functional groups. As lignins are the most resistant tree components in forests, they would prevent the release of the absorbed elements for longer during the litter degradation. The chemical elements that are more important for tree nutrition and metabolism would then be preferentially released to the soil solutions.

515        As shown by the evolution of the chemical composition of leaves and litter fractions along the different degradation stages, our hypothesis on the distribution of different elements among the different tissues is confirmed. This finding is further corroborated by the result of the leaching experiment, which clearly confirms our hypothesis that during litter decay, the release of elements is linked to the degradation stage of the litter itself. As conjectured, elements partitioned in the most labile tissues are more easily released during the degradation

process than those bound to more refractory tissues, which are instead accumulated over time.

       According to our findings, two main REE fractionation processes are specific to a leaf's life-span:

(i) An inter-tissue fractionation occurring during the leave's "living period", through which recalcitrant tissues would preferentially absorb REE as a result of binding with lignins, developing a particular signature;

(ii) A degradation-driven fractionation, which has the different affinities of REE towards the products of the decay

as the main factor for their partitioning between the remaining solid fraction of litter and the resulting solution.

**4.2 Cerium anomalies in leachates**

Another interesting aspect of our results is the presence of small but significant positive Ce anomalies in the leachates of the oldest litter fractions (Ce/Ce*=1.16 and Ce/Ce*=1.49 in Do OLv and Be OF leachates, respectively) and a slight W-type tetrad effect in the Be OF leachate. The tetrad effect can be defined as a graphical

effect that divides the REE patterns into 4 segments, so-called "tetrads" (T1 from La to Nd; T2 from Pm to Gd; T3 from Gd to Ho; T4 from Er to Lu), resulting from the increased stability at a quarter, half, three-quarter and complete filling of the 4f orbital (McLennan, 1994). The tetrad effect is usually classified according to the shape of the patterns into the "W" type and "M" type.

       Davranche et al. (2005) demonstrated that the REE complexation by organic acids inhibits the

development of the tetrad effect and of Ce anomalies in the REE patterns of aqueous solutions. This is because the complexation operated by organic acids is not selective towards any specific lanthanide and therefore also Ce. The REE complexation with the organic acids can therefore explain the absence of Ce anomalies and of the tetrad effect in the REE patterns of the leachates of the younger litter fractions (Do FL, Do OLn, Be FL, Be OLn, Be OLv), but it does not justify the positive Ce anomalies found in the patterns of the leachates of the oldest fractions

(Do OLv and Be OF) and the W-type tetrad effect observed in the pattern of the Be OF leachate. The presence of both positive Ce anomalies and the tetrad effect can be explained by a biological-driven accumulation of





manganese oxides ($MnO_2$) on the surface of the components of the oldest litter layers and their subsequent transport into solution. Keiluweit et al. (2015) demonstrated how litter decomposition is controlled by the manganese redox cycle. The authors, indeed, explained that during the first three years of the litter degradation,

specific microorganisms (in particular fungi) are able to transform the $Mn^{2+}$ supplied by the decomposing organic material into the more reactive $Mn^{3+}$ form. This latter would be subsequently used by other microorganisms for the degradation of the aromatic compounds (such as lignin and tannins) through redox reactions with the litter components, which would give the Mn back under its reduced $Mn^{2+}$ form. After the first few years, the excess of $Mn^{3+}$ produced by the biological activity precipitates under the form of $Mn^{3+}$ / $Mn^{4+}$ oxides accumulating on the

surface of the litter during more advanced stages of degradation (Keiluweit et al., 2015).

Unlike organic acids, manganese oxides are capable of a selective adsorption of Ce, along the other REE, with a mechanism of oxidative scavenging through which Ce is preferentially trapped onto the surface of the above-mentioned oxides (Bau, 1999; Bau and Koschinsky, 2009, Pourret and Davranche, 2013). The Ce enrichment linked to Mn oxides could be the reason for the formation of positive Ce anomalies in the waters that

leached the litter material.

We conjecture that after a rainfall event, residual water that is deposited onto the surface of the oldest litter fraction has inherited a specific REE signature after passing through the younger litter layers above. We can assume that such a signature is similar to that of the leachates of the younger litter fractions recovered during our experiment (with the related MREE-HREE enrichment). Once the $MnO_2$ deposited onto the litter surface interacts

with this solution, it would preferentially adsorb Ce with the scavenging mechanism previously mentioned. A question mark here is related to the form (complexed or free ions) of the REE when they enter into contact with the $MnO_2$. We assume that their main form during such an interaction occurs mainly as free ions as their complexation with organic acids could inhibit the preferential adsorption of Ce onto $MnO_2$ as observed by Davranche *et al.* (2005 and 2008). They also highlighted a process of REE-organic acids complexes dissociating

with time and with decreasing $HA/MnO_2$ ratios. The reduced DOC concentrations (Table SI-4) and the presence of the $MnO_2$ in Do OLv and Be OF would then lead to the dissociation of the OA-REE complexes and to the re-adsorption of the REE onto the $MnO_2$ with a preferential intake of Ce. Note that when compared to the Do OLv leachate, the Be OF leachate shows lower DOC, a higher Ce anomaly and the presence of TE, which may be a direct effect of the decrease on the $OA/MnO_2$ ratios on the development of these specific REE features in the

solutions during the litter degradation.

Interestingly, for both species the leachates of younger litter fractions (Do FL, Do OLn, Be FL, Be OLn and Be OLv) show higher DOC concentrations and lower pH than those of the oldest litter fractions (Do OLv and Be OF), as shown in Table SI-4. This strengthens the assumption that the REE patterns in leachates of fresh leaves and young litter fractions are shaped by the presence of organic acids, which confers the typical increasing trend

from La to Lu and the absence of positive Ce/Ce* and of TE (Fig. 4). On the contrary, leachates of the oldest litter fractions show higher pH, lower DOC, positive Ce anomalies and TE (in Be OF leachate), indicating that the shapes of the REE patterns in the leachates of the oldest litter fractions are mainly a result of OA-REE dissociation accompanied by Ce-enriched $MnO_2$. This would explain both the increasing trend from La to Lu and the development of positive Ce anomalies (with a TE in the Be OF sample) in the leachates of the oldest litter fractions.

We propose that the process leading to Ce enrichment in waters that are in contact with the oldest litter fractions occurs in three steps, as reported below (and more accurately in Figure 5):



    i.   Biologically-driven accumulation of $MnO_2$ particles onto the surface of the old litter components;

    ii.   Dissociation of OA-REE complexes and subsequent oxidative scavenging of Ce onto the $MnO_2$ particles' surface in the presence of stationary water in the litter surface during the degradation;

iii.   Dissolution of Ce-enriched $MnO_2$ particles and / or their direct transport as $MnO_2$ nanoparticles into solution operated by incident waters characterized by higher water volume and higher turbulence, which may then "wash" the surface of the oldest litter layers. One or the combination of these two processes would lead to Ce enrichment in the solutions, thus developing a positive anomaly.

One may argue that the yields of Mn during the leaching experiment are higher in the Do OLn and Be OLv

leachates where the Ce anomalies do not appear. Here it is important to consider not the overall concentration of Mn in the leachates, but rather the chemical form in which this element is present in the litter layers. We recall, in fact, that the formation of Mn oxides (which lead to the development of Ce enrichment) only occur during the last stages of the litter decay, in our case Do OLv and Be OF.










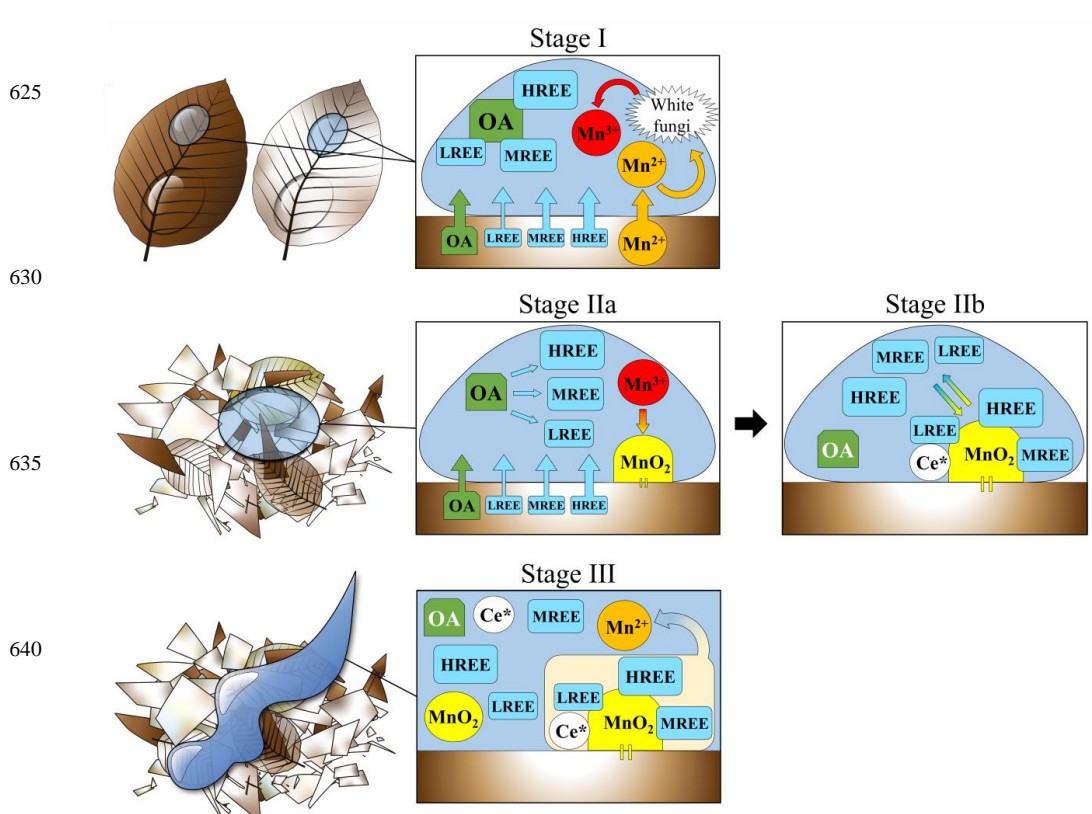

**Figure 5.** Conceptual model representing the main processes treated in this study occurring during the litter degradation.

**Stage I** Do OLn, Be OLn and Be OLv degradation
- Preferential release of HREE and MREE linked to their affinity for organic acids
- REE complexation with organic acids and subsequent inhibition of Ce anomalies and Tetrad Effect
- Transformation of $Mn^{2+}$ (coming from the litter) into $Mn^{3+}$ operated by white fungi

**Stage IIa** Do OLv and Be OF degradation
- Preferential release of HREE and MREE linked to their affinity for organic acids
- REE complexation with organic acids and subsequent inhibition of Ce anomalies and Tetrad Effect
- Accumulation and precipitation of $Mn^{3+}$ under the form of Mn oxides
- Decrease in the $OA/MnO_2$ ratios (due to Increased concentration of $MnO_2$ and decreased release of organic acids), lead to the REE-organic acids dissociation

**Stage IIb** Do OLv and Be OF degradation
- REE released from the organic acids in **Stage IIa** are re-adsorbed onto the $MnO_2$ particles positioned on the litter surface
- Scavenging of Ce and its subsequential enrichment on the Mn oxide surface

**Stage III** Do OLv and Be OF degradation
- Higher volume of water and higher turbulence lead to the dissolution and/or direct transportation of the Ce-enriched Mn oxides particles into solution, which inherits the enrichment in Ce and develops a tetrad effect (Be OF only)



### 4.3 Behaviour of Ca and Eu during litter degradation

Ca shows percentages of leaching that are close to or even lower than trivalent metals during the first stages of Douglas-fir leaf degradation (Do FL and Do OLn), while its release increases from the oldest litter layer (Do OLv). In European beech samples, the leaching of Ca is not as low as from Douglas-fir samples but is lower than the other divalent elements (Mn and Mg). At high concentrations, Ca is a toxic element for trees. Turpault *et al.* (2021 and reference therein) argued that during the leaves' senescence, Ca can move from the tree's body to the leaves where it crystalizes under the form of insoluble Ca-bearing bio-minerals (such as calcium oxalate), as a form of anti-toxicity mechanism. Calcium is also involved in other plant mechanisms; among these, the stabilization of the cell wall structure is of vital importance. Indeed, it is an essential component of the calcium pectate, an insoluble molecule that forms polymers in between the cell walls linking them together (Bateman and Basham, 1976; Proseus and Boyer, 2012). The fact that calcium pectate is insoluble and that is positioned in between the cell walls makes this molecule less accessible to microorganisms during the initial stages of the degradation, leading to a reduced release of Ca. The fragmentation of the leaves and the decay of the weakest tissues during the first stages of degradation would therefore facilitate the accessibility of these insoluble components to the biological degradation (Norman, 1929), thus contributing to the calcium release from the oldest litter fractions.

Noticeable is also the increase in the Eu/Eu* values in Douglas-fir samples, passing from 0.96 in Do FL to 1.09 in Do OLn. We are aware that a value of 1.09 does not represent a real anomaly but this increase, followed by a decrease in the Do OLv fraction (Eu/Eu* = 0.93) may help to understand the Ca behaviour. $Eu^{3+}$ has already been shown to be able to replace $Ca^{2+}$ in some physiological mechanisms due to similarities in their ionic radii (Amann et al., 1992; Zeng et al., 2003; Shtangeeva and Ayrault, 2007). Therefore, the Eu/Eu* increased value in Do OLn may be due to the partial involvement of Eu in the Ca-dedicated biochemical pathways described above, which would lead to an anomalous accumulation of Eu in respect to the other REE in Douglas-fir litter. The fact that the increase in the Eu/Eu* value has been found in Do OLn and not in Do FL strengthens the assumption that Eu is involved in the same process of translocation of Ca that occurs during the leaves' senescence described by Turpault *et al*. (2021) and previously reported in this chapter. Although the reason for calcium behaving differently in the two tree species during the litter degradation cannot be explained with our experiment, it could be due to possible differences in the chemical and/or mechanical structures of the leaves or in the physiology of the species.

### 4.4 Rare Earth Elements as a proxy for litter degradation resistance?

Both tree species show a progressive decrease in the Y/Ho ratios, indicating that during the degradation of the litter material, Y decouples from Ho, as it is preferentially leached. This trend in the Y/Ho ratios is also accompanied by the enrichment in LREE proceeding towards decay in both species, as shown by the $La_N/Yb_N$ ratios. These ratios can be thought of as proxies for classifying resistance to the litter degradation of the two tree species in the Weierbach forest. As illustrated in Figure 6, the smaller the slope of the regression line, the lower the resistance. In accordance with this, Douglas-fir samples appear to be less resistant than the European beech ones. This is in line with the higher yields of trivalent metals we observed in the leachates of Douglas-fir samples as they are bound to the most resistant tissues. It is interesting to note that the fresh leaves from both species have a close position in the graph, indicating a similar stage of degradation (none) which changes over time.





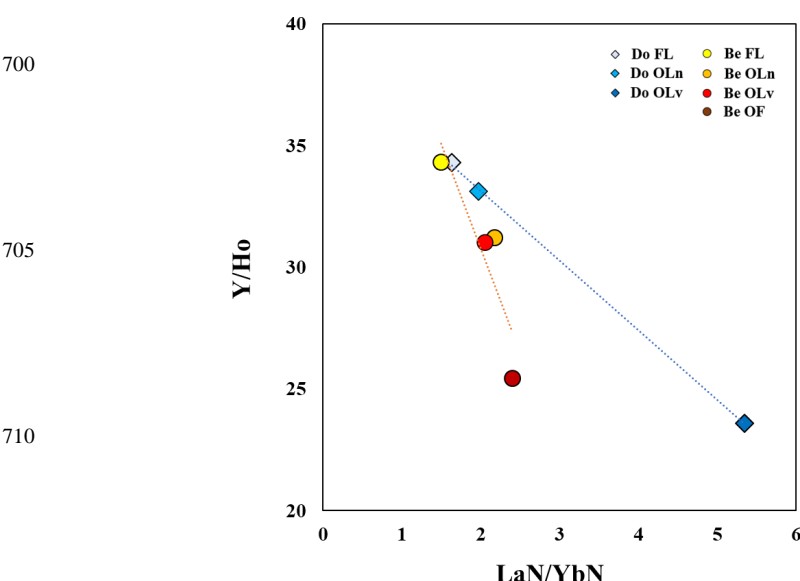

**Figure 6.** Y/Ho vs LaN/TbN ratios of fresh leaves and litter samples of Douglas-fir (blue scale) and European beech (red scale).

**4.5 REE in soil solutions**

The differences in the average soil solution REE patterns observed between the two experimental sites in the

Weierbach catchment seem to be linked to the different REE release occurring during the degradation of the litter at each plot. Indeed, from a depth close to the litter layers (soil solution at 20 cm depth) to the deepest soil layer (soil solution at 60 cm depth), the evolution of the HREE enrichment, Ce anomaly and specific MREE (Gd and/or Eu) enrichments in soil solutions (Fig. 7) could be discussed according to similarities with the litter leachates (Fig. 3 c-d). It may be expected that if any litter degradation compounds can contribute to the soil

solution REE composition, it would be more easily observed close to the surface and would disappear progressively with depth, being diluted by the water-rock interaction processes and changing in redox conditions that control REE in soils (Braun et al., 1998; Laveuf and Cornu, 2009). Our results are in accordance with this expectation. For instance, particularities in the REE patterns of these litter leachates (especially for the last two stages of degradations for both species) seem to be mirrored by their respective soil solutions. Indeed, Eu and

Gd enrichments, observed independently in both litter leachates (Do OLv and Be OF), were also found in the related soil solutions ($1.08 \leq Gd/Gd^* \leq 1.21$ in Be SS and $1.06 \leq Eu/Eu^* \leq 1.15$ in Do SS). In both profiles, such anomalies are higher at 40 cm and decrease at 60 cm. This is in line with the leachate REE patterns normalized by the respective bulk litter concentrations reported in Figure 4. Indeed, in the patterns of Be OLv and Be OF, Gd is the most concentrated, while in patterns of Do OLn and Do OLv Eu leads the MREE enrichment. This

illustrates a preferential release of these two elements to the soil solutions during the natural leaching operated by rainfall and throughfall on Do OLn, Do OLv fractions in the Douglas-fir stand and of Be OLv, Be OF fractions in the beech stand.





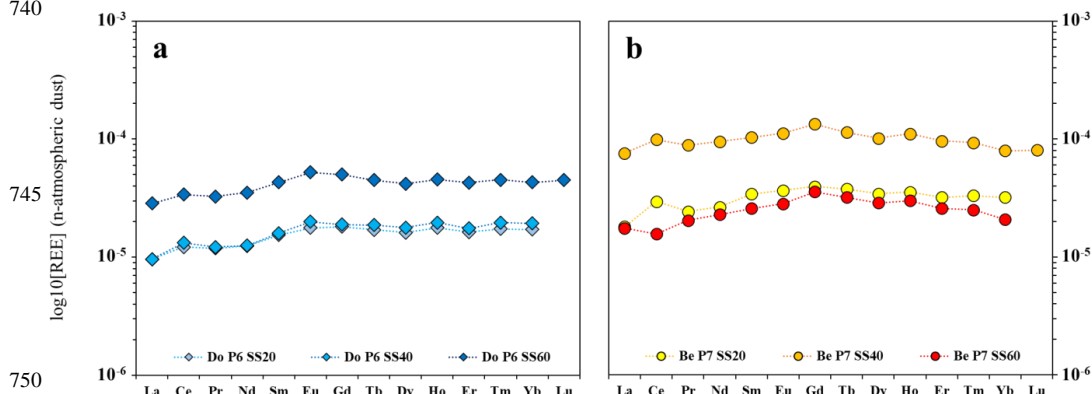

**Figure 7**. Patterns of the average Rare Earth Elements concentrations in soil solutions of DoP (**a**) and BeP (**b**). REE concentrations have been normalized by the values in the local atmospheric dust.

Moreover, REE patterns of soil solutions from both profiles show Ce positive anomalies (Ce/Ce*=1.14 in Do SS20 and Ce/Ce*= 1.39 in Be SS20) that are close to those in the leachates of the oldest litter fractions. The amplitude of these anomalies decreases with the increasing depth until they disappear at 60 cm. Here again, the natural leaching of the oldest litter material could lead to the desorption of REE from the $MnO_2$ deposited there or the direct transport into solution of Mn oxide nano-particles enriched in Ce, finally leading to a positive Ce anomaly in soil solution. However, due to its redox sensitive nature, Ce dynamics are not easily understandable in soil solutions in which, due to the oxidative conditions we would expect there to be a depletion (negative Ce/Ce*) due to the precipitation of $Ce^{4+}$ as cerianite and adsorption onto Fe-Mn oxy-hydroxides.

Other important features of leachate patterns that are mirrored in the soil solutions are the HREE and MREE enrichments for the Douglas-fir and European beech stands, respectively, when compared to LREE. Indeed, average Douglas-fir soil solutions showed quite stable $La_N/Yb_N$ ratios at all depths with values comprised between 0.50 (Do SS40) and 0.66 (Do SS60), which are in line with the $La_N/Yb_N$ ratios of Douglas-fir litter leachates ($0.10 \leq La_N/Yb_N \leq 0.82$). Concerning the beech stand, $La_N/Gd_N$ in average soil solutions shows values between 0.46 (Be SS20) and 0.57 (Be SS40), still in line with the values of beech litter leachates ($0.43 \leq La_N/Gd_N \leq 0.51$) obtained with our experiment. Similarities like these suggest a strong impact of the litter degradation on what is the REE signature of soil solutions, especially at shallower depths, and the fact that the anomalies tend to disappear in the deepest solutions strengthen this assumption.

It must be said that the same environmental conditions to which the litter is generally exposed are not found in the laboratory. Conditions under which a greater degradation efficiency would be expected were avoided due to limitations present in the laboratory (such as the limited exchange of gases with the atmosphere, limited light, greater volume of water per litter surface area, lower concentration of microorganisms). Additional *in-situ* studies regarding the REE dynamics in the Weierbach catchment's soils are necessary to better understand and quantify the real contribution of litter degradation to the REE composition of soil solutions in a forest ecosystem.



Moreover, chromatographic analysis of the leachates and SEM analysis of litter surfaces could help, respectively, to elucidate what kind of REE ligands are present in the different leachates and to observe the existence (or not)

of $MnO_2$ particles deposited on the surface of the oldest litter fractions.

## 5 Conclusions

We focused our attention on the role of forest vegetation on REE and the associated sequestration and release of major cations into and from leaf tissues during the litter degradation. As shown in our experiment and similarly for both tree species, major cations and nutrients like Na, Mg, K, and Mn are preferentially located in more labile

tissues and are easily released during litter degradation, while Pb, Fe, Al and REE tend to be accumulated in the most recalcitrant tissues. We conjecture that such a sequestration in degradation-resistant tissues is imputable to the binding with lignins as the most resistant compounds in leaves.

Our results clearly show that litter degradation plays an important role in the REE dynamics in forest ecosystems. New findings related to REE dynamics during litter degradation and the potential of REE as

complementary tracers for litter degradation processes were highlighted. The observation that the Ce anomaly and tetrad effect only occur in leachates of the oldest litter fractions can be linked to the accumulation of $MnO_2$ on the surface of the litter after the first years of degradation. In comparison to the major cations, REE presented significant differences during the degradation of the litters of European beech and Douglas-fir. In this latter, Eu seems to be involved in the same Ca translocation pathway that occurs during the leaf senescence. Moreover,

the evolution of the $La_N/Yb_N$ and Y/Ho ratios could be used as a proxy to analyse the resistance to the degradation of the leaves and litter between these two tree species.

Finally, the type of tree cover and the degradation stage of the litter are important parameters to consider when studying the chemistry of REE in forest soil waters. Similarities between the REE patterns of fresh leaves and litter leachates and REE patterns of soil solutions have been reported, possibly suggesting the importance of

vegetation in determining the REE signatures in soil solutions. When compared to the other elements in the series, HREE are preferentially released from litter into solution due to the stronger affinity they have with the organic acids produced during the leaves' degradation stages. This would also explain the unexpected positive Ce anomaly that can be observed in the shallower soil solutions of the Weierbach experimental catchment in Luxembourg.


## 6 Data availability

The database used in this study is publicly available at zenodo.org (https://doi.org/10.5281/zenodo.5569559).

## 7 Acknowledgements

This work is part of the HYDRO-CSI project and was supported by the Luxembourg National Research Fund

(FNR) in the framework of the FNR/PRIDE research programme (contract no. PRIDE15/10623093/HYDRO-CSI/Pfister). We would like to thank Lindsey Auguin, for the English proofreading of the manuscript.





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
