# Peer review of "Dynamics of Rare Earth Elements and associated major and trace elements during Douglas-fir (*Pseudotsuga menziesii*) and European beech (*Fagus sylvatica* L.) litter degradation."

_Biogeosciences, 2021_

## Author Response (AR1)

Biogeosciences Discuss., author comment AC1
https://doi.org/10.5194/bg-2021-268-AC1, 2021

[Figure]

**Reply on RC1**

Alessandro Montemagno et al.

Author comment on "Dynamics of Rare Earths and associated major and trace elements during Douglas-fir (*Pseudotsuga menziesii*) and European beech (*Fagus sylvatica* L.) litter degradation" by Alessandro Montemagno et al., Biogeosciences Discuss., https://doi.org/10.5194/bg-2021-268-AC1, 2021

Dear Dr. Pourret,

first of all I would like to thank you, on behalf of all the authors, for the time spent on reviewing our manuscript and for the useful insights you took care to share with us.

In the same order of your questions/comments, please find below our corresponding answers:

*- Analytical section (sample analysis) is very short, details should be added especially regarding QA/QC; isobaric interferences may occur, it must be checked and clearly stated.*

The concentrations of major cations and trace elements in all samples were analysed via Inductively Coupled Plasma - Mass Spectrometry (Agilent 7900). The measured isotopes were chosen with no isobaric interferences, and the polyatomic interferences were minimized by using the collision cell in Helium mode. Calibration standards and Quality Controls (QCs) were prepared with certified solutions (Chem-lab, Belgium and Merck, Belgium). QCs at low, medium and high levels of concentration of the calibration range were analysed each after ten samples to control the validity of the measurement. The limits of quantification (LoQ) for the different analysed elements are reported in Tables SI-3 and SI-4. Also, mineralization blanks were prepared following the same steps as for the samples to ensure the quality of the procedure.

*- In the discussion section, it would be usefull to check how microorganisms, siderophores,... may have a role on redox behavior of Ce. It is stated in the M&M section that biotic communities were preserved.*

It is true that microorganisms may affect the redox behaviour of Ce through specific mechanisms. Unfortunately, we did not investigate the composition of the biotic communities in our samples but, among the microorganisms that have been found to have a potential role in the REE mobilization and behaviour, we proposed the ones fitting the best our data in order to develop the conceptual model. Initially we did not think about the role of siderophores in such a system and we thank you for the useful suggestion. Nonetheless, some considerations can be made regarding these specific compounds and their potential role in our experiment. The release of siderophores may occur both from

fungi and bacteria to help the Iron assimilation, by the way we cannot imagine these compounds to play an important role in the redox behaviour of Ce in our system. Indeed, the release of siderophores is more a condition-specific mechanism (it occurs in iron-deficiency conditions – Chennappa et al. 2019 https://www.sciencedirect.com/science/article/pii/B9780444641915000195) rather than timeframe-specific (e.g. during specific periods of litter degradation). Thus, we would expect sideriphores to be released to enhance the Iron assimilation especially during the first stages of the degradation where the Fe concentrations are much lower when compared to the oldest fractions. Therefore, it is precisely in those early stages of decay where one would expect a greater participation of the siderophores in the geochemical behavior of Cerium. Kraemer et al. (2017 - https://www.sciencedirect.com/science/article/abs/pii/S0016703716305336) demonstrated that siderophores are able to scavenge Ce by oxidizing it to Ce (IV) forming stable complexes and leading to the development of positive cerium anomalies in solution, anomalies that are not shown in the leachates of the younger litter fractions where, as mentioned before, we would expect them according to the Fe-deficiency. It is important, therefore, to remember that the Ce enrichment occurs only in the leachates of the oldest litter fractions, which suggest that the process acting in these circumstances is timeframe-specific and occurs during the latest stage of the degradation. Other important aspects to take into account are the much higher concentrations of manganese in the leachates when compared to iron (Table SI-4) and the different behaviour (in terms of % of yields) that this latter shows between the leachates of the two tree species, while Ce instead shows the same dynamics in terms of anomaly development.

*- In the same part, authors consider Mn oxides, what about Fe oxides?*

The competitive scavenging of Ce operated by Fe and Mn oxides is a quite controversial argument. By the way, Bau and Koschinsky (2009 - https://www.jstage.jst.go.jp/article/geochemj/43/1/43_1.0005/_article) demonstrated, through a sequential leaching experiment carried on ground Fe-Mn crust samples, that (in marine environment) also Fe oxides are able to scavenge Ce from the water column with a lesser extent when compared to Mn oxides as demonstrated by the higher positive Ce/Ce* found in the Mn oxides fraction. Despite the fact that they found similar Ce/Ce* anomalies in the two oxides, we have to mention that the Mn/Fe ratios in our leachates ($33.6 \leq$ Mn/Fe $\leq 276.5$) are much higher than the ones of the total concentrations in the samples treated by the above-mentioned authors ($1.57 \leq$ Mn/Fe $\leq 1.65$). This suggests that Mn is acting as protagonist in the Ce oxidation in our system rather than Fe. Of course our experiment is not a marine environment replica, but it is also true that our is the first attempt to explain not only Ce behaviour but more in general the REE dynamics during the litter degradation and this makes difficult to find suitable literature to compare our results with.

*-In the discussion section, the part dealing with Eu anomaly need to consider all the relevant literature on the subject, check my latest article https://doi.org/10.1007/s11104-021-05210-6...*

We agree with your comment and we also believe that this literature should be implemented in our manuscript in order to give a broader overview on Eu behaviour in plants.

As you discussed, plenty of processes may result in the mobilization, sequestration and fractionation of REE in plants tissues. Among all these processes, the mobilization of Eu to the leaves from other tree's organs before the leaves' senescence is the one which may explain the (small) Eu enrichment we observe only in the new litter of Douglas-fir (Do OLn, Eu/Eu* = 1.09) which represent the newly deposited leaves just after the senescence. Indeed, If other processes (such the ones you mentioned) played an

important role in the preferential Eu accumulation in our leaves samples, they would have occurred during all the living period and we would have observed an Eu enrichment already in the fresh leaves, which instead does not occur (Eu/Eu* = 0.93 in Do FL). For what it may concern the Eu anomaly in the leachate, we proposed the Eu substitution of Ca in the Ca pectate to justify the positive Eu/Eu* because this latter is occurring only in the leachate of the oldest Douglas-fir litter (Do OLv). Calcium pectate links cell walls giving rigidity to the whole structure. Such a strategic positioning between degradation resistant compounds (cell walls are composed of the most resistant molecules such as cellulose, hemicellulose and lignins), might make this molecule not easily accessible during the early stage of the degradation. This could give a clue on why we observe such an anomaly only in the latest stage of the degradation after that most of the "weak" tissues have been already degraded. Of course, the similarity between the ionc radii of Eu(III) and Ca might play a role on this substitution.

In the case of the other accumulation processes such as Eu binding to organelles and/or inner membranes, or binding with phosphates, we might expect these molecular associations to be among the first cell components to be degraded and/or released due to their chemical composition (organelles and membranes are mainly composed of lipids, proteins, glycolipids and glycoproteins) and position (both organelles and phosphate are present in the cytosol). Thus, in this case, we would have expected the Eu anomaly to appear already in the leachates of the younger litter fraction and fresh leaves. Furthermore, with an in-vitro experiment with Brassica napus cells, Moll et al. (2020 - https://link.springer.com/article/10.1007/s11356-020-09525-2) demonstrated that in plant's cell Eu is preferentially absorbed into cell walls as result of anti toxicity mechanisms. In case of the solely absorption operated by the cell walls, we have no reason to believe that such a mechanism would act preferentially on Eu rather than the other REE (into specific the two Eu neighbours). The enriched release of Eu during the last stage of degradation, instead, is suggesting that at least part of this Eu is fractionated or into slightly less degradation-resistant compounds (when compared to lignins where the other REE are supposed to be bound) or into specific not-easily-accessible positions which make its release into the environment not immediate at the beginning of the degradation. The late degradation of these afore-mentioned compartments would deliver additional Eu content during the degradation of the oldest litter fraction in addition to the Eu released together with the other REE, leading finally to the slight enrichment in the leachate of the Do OLv sample.

Finally, Eu/Ca ratios in the first 60 cm of soil and bedrock under the Douglas-fir stand ranged from 0.0005 to 0.0044 (Moragues-Quiroga et al., 2017 - https://doi.org/10.1016/j.catena.2016.09.015) which, according to Brioschi et al. (2013 - https://link.springer.com/article/10.1007%2Fs11104-012-1407-0), for instance, indicate a Ca depletion in the regolith of our experimental site. The question on how such a Ca depletion could play a role on the Eu positive anomaly due to the fact that this latter is shown enriched in the leachates of the oldest litter fraction still remains. For what it may concern instead the slight enrichment in Eu occurring in the solid fraction, as previously mentioned, it is occurring in the Do OLn fraction (which was already on the ground) and not in the fresh leaves where instead the Eu/Eu* is below 1. It is our opinion then, that the enrichment in the leaves is occurring just during the leaves senescence before the falling and not during their living period. But we agree on the fact that this question still needs to be studied and that our results cannot entirely answer to it.

-Minor comments

We agree with all the comments of this session and, accordingly, corrections will be applied to the final document. Figures' features have been already checked during the preliminary session for the article submission.

I would like to thank you again and to wish you all the best,

Alessandro Montemagno

[Figure]

Biogeosciences Discuss., author comment AC2
https://doi.org/10.5194/bg-2021-268-AC2, 2022

[Figure]

**Reply on RC2**

Alessandro Montemagno et al.

Author comment on "Dynamics of Rare Earths and associated major and trace elements during Douglas-fir (*Pseudotsuga menziesii*) and European beech (*Fagus sylvatica* L.) litter degradation" by Alessandro Montemagno et al., Biogeosciences Discuss., https://doi.org/10.5194/bg-2021-268-AC2, 2022

Dear Reviewer,

Many thanks, on behalf of all the authors, for accepting to revise our manuscript and for the clever suggestions you provide with such an evaluation.
Please, find below the answers to your questions/comments in the same order as you submitted.

*General comments*
*• The study aims to use the chemical signature of REE to better understand the decomposition rate of plant litter.*
*• I highly recommend normalizing the REE by the lower soil horizon or parent material or continental crust values. It is unusual to normalize by the dust since dust is typically a potential end member. All REE patterns should be reprocessed using a "true" regolith source. Dust can be transported from long distances, how can this be considered the sole parent material?*

It is not usual from a literature point of view, to normalize REE concentrations to dust. Nonetheless it must be said that the normalization by a specific material can usually be done according to given processes to be highlighted (for example Stille et al. 2009 - https://doi.org/10.1016/j.chemgeo.2009.03.005 - who normalised soil leachates to soil solutions). As explained in lines 217-220, atmospheric deposition is an important input of cations and nutrients and we expected it to be also in terms of REE supply as previously suggested by Censi et al. (2017- https://doi.org/10.1016/j.chemosphere.2016.11.085). Therefore, this normalization is (also) a way to observe whether or not the atmospheric deposition has an impact on the REE composition (and thus on patterns) of leaves and litter. Indeed, if we obtained flat litter REE patterns when normalizing to the dust, we could have argued that most of the REE in the solid fraction of litter was delivered by the dust and not by the litter itself.
We remind also that the Weierbach regolith has multiple origins and it is composed of a Devonian slate substrate covered by 70-100 cm of Pleistocene Periglacial Slope aeolian deposit. This would also make difficult the choice of the regolith material to be used as standard for the normalization.
However, by normalising the dust REE concentrations by the PAAS (McLennan 1989 - https://doi.org/10.1515/9781501509032-010), it is possible to notice that the pattern

obtained in this way has almost a flat shape characterised by a slight MREE enrichment ($LaN/GdN=0.88$ and $GdN/YbN=1.16$) and a slight Eu positive anomaly ($Eu/Eu^*=1.12$). Consequently, changing the normalization to the PAAS would then produce almost identical patterns in the litter and in the leachates with the sole difference of an accentuation of the already shown europium anomaly and MREE enrichments.

*Abstract*
*• Write the abstract from the perspective of how the REE signatures inform the biogeochemical process involved on litter degradation. A more compelling story is the use of REE signatures as biogeochemical indicators of degradation steps. The abstract needs to be flipped from a different perspective. Let's say, I collect a sample from the O horizon and determine the REE signature, will this info reveal the stage of the organic matter decomposition. This is how reactive tracers are useful in future applications.*

We propose to improve the abstract according to the following comments.

*• Is there any difference on biogeochemical process under the two different type of litter? Are the REE signatures reflecting these differences? This was not mentioned in the abstract; however, since it is part of the tittle, I believe it should be relevant.*

We propose to add the following arguments to the final abstract:

- According to the REE dynamics, litters of the two tree species showed similar biogeochemical processes dominating these elements behaviour.

- Using REE, two main elements of distinction between the two species were highlighted:

  • Eu behaviour linked to the Ca during the leaves' senescence in the Pseudotsuga menziesii, linkage which was not found in the Fagus sylvatica;

  • Species-specific release of organic acids during the litter degradation which lead to differences in the MREE enrichment of the litter leachates.

*Lines 29-31: This type of statement needs to include the trend. What does a high or low ratio inform the degradation intensity?*

We agree. We also believe that the use of the Y/Ho ratios should be integrated in the statement of lines 29-31 (which refers only to the LaN/YbN ratios), in order to provide a more precise view on how these elements were used to gather information about the intensity of the degradation. We propose to modify the lines 29-31 as follow:

"In particular, the degradation of the litter was characterised by a decrease in the Y/Ho ratio and an increase in the LaN/YbN ratio during the decay. The relationship between these ratios delivered information on the litter species-specific resistance to degradation, with Douglas-fir litter material showing a lower resistance."

*Lines 31-33: The important implication here is how the white fungi activity influences organic matter degradation resulting in the Ce anomaly.*

We agree and, accordingly, we propose to reformulate as follow:

"Finally, we showed the primary control effect that white fungi may have in the Ce enrichment of soil solutions, which appears to be associated with the dissolution and/or direct transport of Ce-enriched $MnO_2$ particles accumulated on the surface of the old litter due to the metabolic functioning of these microorganisms."

*Lines 69-72: Add a reference citation.*

Regarding the fractionation of REE in plants: Liang *et al*. 2008
(https://doi.org/10.1016/S1002-0721(08)60027-7).
The references regarding REE as emerging pollutants are reported in lines 74-75 as the sentence in those lines is connected to the previous one. To make it more clear for the reader, we propose to put those references at the end of the sentence in the lines you suggested.

*Methods*
*• In Figure 1, the authors refer as humus material the fragmented litter. This is an incorrect definition of humus material. The thick brown or black substance that remains after most of the organic litter has decomposed is called humus. Humus material should be unrecognizable which is not the case in Figure 1. This material should be simply called partially fragmented litter not humus.*

We agree on this comment. Subsequently, the correction on the nomenclature of the fragmented litter will be applied to the final document.

*• Please justify why the digestion for the litter was different from the dust.*

We applied the HNO3/HF/HClO4 acid mixture digestion using hot plate in order to totally dissolve the dust sample because this is the standard protocol used in our laboratory for this type of sample. Indeed, HF is needed for breaking the strong Si-O bonds of the silicate mineral phases that could contribute to atmospheric dust (Lequy et al., 2012 in Forest Ecology and Management). Due to the number of samples and the related amount of litter material to be digested, we opted for a more convenient digestion of the organic-derived samples by using a microwave-assisted oven, which allows reaching higher temperature and pressure conditions (maximum temperature and pressure reached 212 °C and 24.2 bar respectively). The microwave-assisted oven technique does not allow the use HF in our laboratory and we decided to test different digestion methods (HNO3/H2O2, Aqua Regia) to validate the most efficient one. Aqua Regia delivered solutions without any precipitates nor suspended particles as they resulted clear and transparent not only at the moment of the digestion but also long time after.
However, we agree that Aqua Regia is not a total digestion method, and this will be specified in the manuscript for clarification.

*What is the scientific rational to use the local atmospheric deposition for normalization? Atmospheric dust should be treated as an end member. The dust can affect the REE signature. Is this normalization widely accepted in the literature? I highly recommend normalizing by the lower soil horizon or parent material or continental crust values. I question if the REE signatures are truly reflecting the decomposition of the litter or are biased by the dust normalization.*

Please, for this point, refers to the first comment.

*Line 180: Aqua Regia is not a total digestion method. Please clarify that this is a partial digestion.*

We agree and will clarify that aqua regia is not a total digestion method as you suggest.

*Section 2.3: Include information about the standards used in the calibration curve and the internal standard.*

The calibration standards were prepared with Chem-lab (Belgium) and Merck (Belgium) certified solutions, while the internal standards were prepared with Chem-lab (Belgium)

rhenium and rhodium certified solutions.

The REE groups were already defined in lines 76-77

*Results*
*Line 335: Specify, results are for the solid matrices.*

It will be done.

*Line 336: significant different… was this tested statistically? If not, how do you know is significant different?*

We noticed that the use of the word "significant" was not appropriate in this manuscript. In this particular case, the intention was to draw attention to specific aspects of the REE pattern shapes to highlight the differences between the patterns of the two tree species. We propose to remove the word "significant" in this instance.

*Line 366/374: significant Eu/Ce positive anomalies… where are the statistics to support these statements?*

Also in this case, "significant" was not related to the use of statistics. Just an inappropriate use of the term, which was instead intended as "meaningful". Again, we propose to remove the word "significant".

*Discussion*
*It is confusing in some instances to follow if the discussion relates to the solid or leachates. Please make it clear across.*

We will proceed to specify whether the sentences refer to the solid fraction or to the leachates.

*Lines 425-433: This reads as results descriptions. There are no implications associated to these descriptions.*

We wanted to use those lines as an introduction to the next sentences of the manuscript. We suggest removing lines 426-433 and improving the associated discussion in the following text.

*Line 527: … significant positive Ce anomalies… Where is the p-value?*

As above, we propose to remove the word "significant" in order to avoid these misunderstandings.

I would like to thank you again and to wish you all the best,

Alessandro Montemagno

**Dynamics of Rare Earth Elements and associated major and trace elements during Douglas-fir (*Pseudotsuga menziesii*) and European beech (*Fagus sylvatica* L.) litter degradation.**

Relevant changes according to the RC suggestions have been applied to:

**- Abstract** according to RC2 suggestions: partly rephrased and harmonized with a focus on the processes highlighted in the manuscript;

**- 4.2 Cerium anomalies in leachates:** considered all the relevant literature on the subject as suggested by RC1, then rephrased (partly) and harmonized accordingly;

**- 4.3 Behaviour of Ca and Eu during litter degradation:** considered all the relevant literature on the subject as suggested by RC1, then rephrased (partly) and harmonized accordingly.

---

## Author Response (AR2)

Dear Aninda,

first of all many thanks for your efforts on making our manuscript better.

We would like to reply to your request for a normalization to the soil with some points that we think must be considered:

1) As already replied to reviewer 2, the Weierbach Catchment regolith is a polygenetic system composed of a Pleistocene Periglacial Slope Deposits (PPSD), which was formed by aeolian deposition and in which the soil is developing, and a saprolite deriving from the weathering of a slate bedrock. Here, we do not know from where the trees uptake the nutrients/water and this put constraints on the choice of the most suitable material for the normalization. This is linked due to differences in REE composition between the different regolith layers. Please also refer to the answer to reviewer 2 reported below:

   "It is not usual from a literature point of view, to normalize REE concentrations to dust. Nonetheless it must be said that the normalization by a specific material can usually be done according to given processes to be highlighted (for example Stille et al. 2009 - https://doi.org/10.1016/j.chemgeo.2009.03.005 - who normalised soil leachates to soil solutions). As explained in lines 217-220, atmospheric deposition is an important input of cations and nutrients and we expected it to be also in terms of REE supply as previously suggested by Censi et al. (2017- https://doi.org/10.1016/j.chemosphere.2016.11.085). Therefore, this normalization is (also) a way to observe whether or not the atmospheric deposition has an impact on the REE composition (and thus on patterns) of leaves and litter. Indeed, if we obtained flat litter REE patterns when normalizing to the dust, we could have argued that most of the REE in the solid fraction of litter was delivered by the dust and not by the litter itself.".

2) It is true that plants mobilize and absorb REE from soils. However, preliminary results obtained for this soil proved that only a little fraction of REE are mobilized via organic acids leaching (results not shown in this manuscript). Moreover, non all the REE mobilized in this way are uptaken by trees and part of this "available" REE are removed by hydrological processes, such as percolation.

   Also, the normalization by bulk soil is not representative of what REE are mobilized and uptaken by trees. Indeed during water-rock and organic acids-rock interactions REE fractionate with respect of the bulk soil composition according to the literature. Moreover, it has been demonstrated that also during the root water uptake REE fractionate in roots, changing their concentrations (and patterns) in the xylem.

   Accordingly, we believe that normalizing by the average soil composition would create a bias in the interpretation of the REE patterns of the litter which do not have contact with the below soil (due to our sampling approach) and which during the living period were nourished by water with already fractionated REE signature (xylem water). On the contrary, we expected atmospheric dust to be an active part of the different litter fractions, in terms of REE composition due to its potential direct contact with the collected samples.

3) The addition of a paragraph treating the normalization by soil, we believe would not fit the scope of the manuscript as we do not treat the REE mobilization from the soil particles.

4) Nonetheless, by normalizing to soils, we do not see differences in the REE patterns of our samples. The only noticeable difference is related to an emphasis of Eu anomaly in Do OLn already observed and treated in the manuscript (please see pictures below reporting the fresh leaves and litter samples normalized by average soil REE composition, you can compare with the figures in the manuscript). The resulting patterns would not then add anything impactful, which may justify an additional paragraph in the manuscript.

[Figure]

*Rare earth elements patterns of fresh leaves and litter samples of Douglas-fir and European beech, normalized to the REE average composition of the local soil.*

Additional corrections applied to the manuscript:

1) Figure 2: fixed a typo in the measuring unit from "ug/g" to "µg/g"
2) Line 610: fixed a typo from "occur" to "occurs"

Best regards,

Alessandro Montemagno